# Early Use of Sotrovimab in Children: A Case Report of an 11-Year-Old Kidney Transplant Recipient Infected with SARS-CoV-2

**DOI:** 10.3390/children9040451

**Published:** 2022-03-23

**Authors:** Costanza Di Chiara, Daniele Mengato, Marica De Pieri, Germana Longo, Elisa Benetti, Francesca Venturini, Carlo Giaquinto, Daniele Donà

**Affiliations:** 1Department for Women’s and Children’s Health, Division of Pediatric Infectious Diseases, University of Padua, 35128 Padua, Italy; marica.depieri@aopd.veneto.it (M.D.P.); carlo.giaquinto@unipd.it (C.G.); daniele.dona@unipd.it (D.D.); 2Hospital Pharmacy Department, University Hospital of Padova, 35121 Padua, Italy; daniele.mengato@aopd.veneto.it (D.M.); francesca.venturini@aopd.veneto.it (F.V.); 3Department for Women’s and Children’s Health, Division of Pediatric Nefrology Unit, University of Padua, 35128 Padua, Italy; germana.longo@aopd.veneto.it (G.L.); elisa.benetti@aopd.veneto.it (E.B.)

**Keywords:** sotrovimab, children, viral-neutralizing monoclonal antibodies, off-label, SARS-CoV-2, COVID-19, pediatric population

## Abstract

Background: The use of virus-neutralizing monoclonal antibodies has been approved in fragile populations, including kidney transplant recipients, who are at risk of developing severe COVID-19. Sotrovimab is the only currently available anti-SARS-CoV-2 neutralizing monoclonal antibody with activity against the new Omicron variant of concern. While sotrovimab has been approved in adolescents and adults, studies regarding its efficacy and safety in children aged less than 12 years old and weighing less than 40 kg are still lacking. Here, we report a first case of a child, who was treated early with sotrovimab after a kidney transplant. Case Report: At the end of January 2022, a 11-year-old male child underwent a deceased-donor kidney transplant and became infected with SARS-CoV-2 during the first day after surgery. Due to the increased risk of developing severe COVID-19, based on the predominance of Omicron and the patient’s renal function, the child was treated with sotrovimab. The clinical course was successful and no adverse reactions were reported. Conclusions: For the first time, we report the well-tolerated use of sotrovimab in children under 12 years old. As the pandemic affects children across the globe, urgent data on sotrovimab dosing in children with a higher risk of developing severe COVID-19 are needed.

## 1. Introduction

Kidney transplant recipients (KTRs) are particularly vulnerable to severe acute respiratory syndrome coronavirus 2 (SARS-CoV-2) infection due to the complex immunosuppressive regimens of transplantation and to the decreased T cell immunity [1,2,3]. Therefore, identifying early interventions that could prevent progression to severe coronavirus disease 2019 (COVID-19) is especially important in this fragile population. Virus-neutralizing monoclonal antibodies (mAbs) are safe and efficient in reducing SARS-CoV-2 viral load in both immunocompetent and immunocompromised outpatients, including solid organ transplant recipients [4,5,6]. Combined with immunosuppression reduction [7,8], mAbs are recommended for solid organ or hematopoietic stem cell transplant recipients with mild to moderate COVID-19 who are at an increased risk of progressing to severe disease [9]. The US Food and Drug Administration (FDA) has issued Emergency Use Authorizations (EUAs) for multiple mAbs therapies for the treatment or postexposure prophylaxis of COVID-19 in patients with certain high-risk conditions [10,11,12]. Similarly, the same have been applied in Europe by the European Medicines Agency (EMA) through the Conditional Marketing Authorization (CMA) procedures [13,14]. While casirivimab + imdevimab and sotrovimab have been approved in adolescents (≥12 years of age) and adults weighing at least 40 kg [10,11], an EUA for the bamlanivimab + etesevimab combination was also authorized by the FDA in children under 12 years of age weighing less than 40 kg [12]. However, the B.1.1.529 (Omicron) variant of concern (VOC), which is now the dominant SARS-CoV-2 variant worldwide, is predicted to have markedly reduced susceptibility to the bamlanivimab + etesevimab and casirivimab + imdevimab mAbs combinations [15]. In fact, due to its numerous mutations in the spike protein, Omicron could escape from certain monoclonal antibodies [16]. Sotrovimab, a recombinant engineered human mABS, is able to bind a highly conserved epitope on the spike protein receptor binding domain (S-RBD) with high affinity (dissociation constant Kd = 0.21 nM) [17]. Moreover, by not competing for binding to the human angiotensin 2 converting enzyme (ACE-2) receptor, sotrovimab, unlike other monoclonals, appears to have a lower risk of ineffectiveness against possible variants of the virus [17].

Although infected pediatric patients have often shown a favorable COVID-19 outcome [18,19], an increased risk for progression to severe disease has been reported in those with comorbidities [20,21]. The significant increase in pediatric COVID-19 cases requires extending the indications for the use of antivirals also to children who are at risk of severe progression of the infection. In light of the growing incidence of COVID-19 in children in the current fourth pandemic wave, Lanari et al. [22] proposed eligibility criteria for the emergency use of mAbs in pediatric patients. However, sotrovimab is the only available anti-SARS-CoV-2 mAbs with activity against the Omicron VOC [11], but studies regarding its efficacy and safety in pediatric patients are still lacking, and no data are available in children aged less than 12 years old and weighing less than 40 kg. Here, we report a case of an 11-year-old child with a mild SARS-CoV-2 infection after kidney transplant, who was treated early with sotrovimab. 

## 2. Case Presentation

At the end of January 2022, during the fourth pandemic wave, an 11-year-old male child of 36 kg, unvaccinated against COVID-19, with end-stage renal disease caused by nephronophthisis was admitted to the Department of Women’s and Children’s Health-University Hospital of Padova (Veneto Region, Italy) and received a deceased-donor kidney transplant. Immunosuppression was induced with basiliximab and methylprednisolone according to the renal association’s clinical practice guidelines [23]. The nasopharyngeal swab (NPS) SARS-CoV-2 RNA polymerase chain reaction performed at the admission returned a negative result. After surgery, the patient was transferred to the pediatric intensive care unit (PICU), and a standard immunosuppressive regimen including tacrolimus, mycophenolate mofetil, and methylprednisolone were immediately administered.

Although he was completely asymptomatic for COVID-19, 1 day after the kidney transplantation, the child tested positive for SARS-CoV-2 in the NPS performed for weekly periodical screening assessed in the PICU. A second NPS obtained 24 h later confirmed the infection, with an increasing viral cycle threshold. The child did not develop symptoms of infection, the laboratory data showed a modestly increased C-reactive protein level (13.62 mg/L), and the chest X-ray showed no pneumonia. As soon as the SARS-CoV-2 infection was confirmed, mycophenolate mofetil was discontinued, the methylprednisolone dose was progressively reduced according to the renal association’s clinical practice guidelines [21], tacrolimus dosing was adjusted in order to reach a trough of 8–10 ng/mL instead of 10–12 ng/mL as recommended in patients without COVID-19, and treatment with mAbs was infused. Considering the COVID-19 severity and the KTR’s renal function (p-creatinine > 400 umol/L), treatment with mAbs was preferred rather than treatment with remdesivir. Considering that the predominance of the Omicron VOC in the Veneto Region had reached 100% at the end of January, viral sequencing was not performed. The patient, once we had obtained authorization for off-label uses from our department and the parents’ consensus, was treated with sotrovimab at a dose of 500 mg in a single intravenous infusion over 30 min. No adverse reactions were reported. Since the KTR remained asymptomatic, 10 days after the COVID-19 diagnosis, the mycophenolate mofetil treatment was resumed. As the child was still positive for SARS-CoV-2 in the NPS, mycophenolate mofetil was reintroduced below dosage as a precaution. The clinical course was successful, with graft function and full recovery in 10 days and no progression of COVID-19. 

## 3. Discussion

Here, we report the management of SARS-CoV-2 infection in a pediatric KTR who became infected immediately after transplant and was treated with sotrovimab. This is the first described case of off-label use of sotrovimab in children under 12 years old to the best of our knowledge. 

SARS-CoV-2 infection after kidney transplant has a higher mortality rate than in the general population [3,24]; however, most of the previous studies were conducted in adults, while data on the pediatric population are limited.

Although Bansal et al. have recently shown that pediatric solid organ transplant recipients develop mild COVID-19 [25], all SARS-CoV-2-positive children enrolled had gone for at least 2 years after the transplant.

Therefore, a gap in knowledge about the course of COVID-19 in the days immediately following the transplant, when immunosuppression is high, still remains.

Discontinuation of early immunomodulant therapy combined with antiviral treatment with mAbs has been shown to have a beneficial effect on reducing severe outcomes in fragile adults affected by mild and moderate COVID-19 [3,26,27]. The Italian Society of Pediatric Infectious Diseases, in a recent position paper, recommended the use of mAbs in asymptomatic/mild COVID-19 children with risk factors for severe disease progression [28]. Bamlanivimab + etesevimab, casirivimab + imdevimab, and sotrovimab are COVID-19 mAbs that have received EUAs from the FDA and EMA for the treatment of patients with mild to moderate COVID-19 who are at a high risk of progression to severe disease [13,14]. Being designed to block the virus’ attachment and entry into human cells, mAbs are directed explicitly against the spike protein of SARS-CoV-2. Therefore, any crucial mutations in the SARS-CoV-2 spike protein confer the virus with the ability to escape neutralization by RBD-specific mAbs [16]. In light of this, due to many mutations concentrated on its spike protein, the Omicron VOC is totally or partially resistant to the bamlanivimab + etesevimab combination, which is with the unique mAbs combination with proven effectiveness in the pediatric population. 

Consequently, given the lack of knowledge on the risk of progression to severe COVID-19 a few days after the kidney transplant, off-label sotrovimab use was proposed, combined with immunosuppressant adjustments. The use of remdesivir in subjects under the age of 18 who do not require supplemental oxygen and who are at an increased risk of developing severe COVID-19 is not yet approved in Europe [29]; furthermore, the patient’s renal failure in the first post-transplant days represented a contraindication for its use. Finding the correct dose is one of the main issues for physicians prescribing off-label biological medication in children due to their peculiar pharmacokinetic and pharmacodynamic profile [30,31]. In the absence of certain pharmacological data on the pediatric patient, some formulas for reconverting the adult dosage are available in the literature if only the patient’s age or weight is available [32,33]. With the data available to us, we decided to readjust Clark’s formula for our patient by adjusting the patient’s reference weight. In the original formula, in fact, the pediatric dose was calculated by multiplying the known adult dosage by the ratio of the patient’s weight to the weight of a standard adult patient (equal to 68 kg). In our case, considering that sotrovimab is administered at the same dosage from 40 kg and above, as required by the data sheet of the drug, we decided to revise the reference weight of the formula precisely to 40 kg [32,33,34]. The final formula used to calculate the dose was as follows:(Patient’s weight divided by 40 kg) × Adult Dose = Pediatric Dosage.

Considering our patient’s weight (36 kg) and the adult dose (500 mg), we obtained a dosage of 450 mg, similar to the recommended dosage in patients over 12 years. The difference between the calculated and standard dosages was only 10%. Taking into account, moreover, the clinical growth chart of our patient, which was substantially superimposable on that of an individual weighing 40 kg and aged 12 years, we decided to keep the dosage of 500 mg unchanged. Thus, we suggest this as a basic calculation of the dose of sotrovimab, taking this elementary revisitation of Clark’s formula into account. In addition to this, we recommend that clinical growth charts should always be considered.

Sotrovimab combined with a temporary reduction in immunosuppression was well tolerated and associated with favorable clinical outcomes in our kidney transplant child diagnosed with mild COVID-19. To date, there are no published reports of children with COVID-19 treated with sotrovimab. 

Data on sotrovimab’s efficacy in adults are encouraging, and it appears to have an excellent risk–benefit profile. The most common adverse events reported in clinical registration studies conducted in adults are mostly related to hypersensitivity to the active substance and are mainly represented by nausea and a rash [11]. The effects have been reported to be short-lived and to resolve within 24 h. More prospective clinical studies should be performed to confirm the efficacy and safety of sotrovimab in the pediatric population. Moreover, the dosage assessment needs further investigation and, above all, adequate pharmacokinetic studies are needed. 

The growth of COVID-19 incidence in children has not yet been followed by an increase in the availability of antiviral drugs for the pediatric population, particularly in Europe, where the EMA did not extend the use of any antivirals (e.g., mAbs or remdesivir) to subjects under 12 years old. Monoclonal antibodies are already approved in pediatric populations with pneumonia comorbidities for the prevention of respiratory tract infections, such as palivizumab, which is a specific monoclonal antibody used for the prevention of respiratory syncytial virus (RSV) infection in preterm infants [35]. In light of this, more rapid approval for the administration of mAbs in fragile young children with COVID-19 is needed. Moreover, future perspectives should evaluate the routinely prophylactic use of mAbs in fragile children before epidemic waves to prevent SARS-CoV-2 infection. 

## 4. Conclusions 

This case highlights a positive outcome for a KTR child with mild SARS-CoV-2 infection treated with sotrovimab. The patient tolerated sotrovimab without complications and with a favorable post-transplant outcome. As the pandemic affects children across the globe, urgent data on sotrovimab dosage in children with a higher risk of developing severe COVID-19 are needed.

## Data Availability

The clinical documents of the current case report are available from the corresponding author on reasonable request.

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
