# Peer review of "Early Use of Sotrovimab in Children: A Case Report of an 11-Year-Old Kidney Transplant Recipient Infected with SARS-CoV-2"

_children, 2022, doi:10.3390/children9040451_

Round 1
Reviewer 1 Report
Di Chiara et al. present the case of a 11-year-old kidney transplant reciepient who recieved off-label Sotrovimab after diagnosis of an asymptomatic SARS-CoV-2 infection.
This case report is highly relevant in the context of intense Omicron spreading and limited therapies against this viral strain. The manuscript is well written but I suggest some revisions.
Introduction :
Lines 52-55 : Too long. Split the sentence
Lines 55-57 : “Although infected pediatric patients have often shown a favorable COVID-19 outcome, in those with comorbidities, an increased risk for progression to severe disease has been reported” --> “Although infected pediatric patients have often shown a favorable COVID-19 outcome, an increased risk for progression to severe disease has been reported in those with comorbidities”
Please add a sentence in the Introduction part to briefly describe the general characteristics of Sotrovimab
Case presentation:
To ensure a full anonymization, the dates must be removed. In fact, we could easily find who is the 11-year-old child who was transplanted in Padova on 30/01/2022. Events should be described as “X days after KTx”. Although, for the relevance of the case report, it is very important to still mention the time period when it happened (fourth wave, predominance of Omicron)
When were the immunosuppressants returned to normal dose?
Was the virus confirmed to be an Omicron ? If yes, was it done before or after the mAbs infusion ?
Discussion :
It would be relevant to mention the adverse events that could happen with sotrovimab
Here, the authors consider that pediatric and adult solid organ transplant recipients have the same higher risk for severe COVID-19. Though, it was reported that pediatric SOT recipients tend to have the same risk for severe COVID-19 than immunocompetent children (for exemple : COVID‐19 infection in pediatric solid organ transplant patients - Bansal - 2022 - Pediatric Transplantation - Wiley Online Library). It should be discussed in this case report
Reviewer 2 Report
The paper called author`s Chiara di C et all. is a case report .The paper presents a good assessment of the author for the use of a new drug in a child on immunosuppressive therapy. The case report is wonderfully handled
Author Response
We thank you for your interest and the valuable comments on our manuscript.